# Road Performance and Self-Healing Property of Bituminous Mixture Containing Urea–Formaldehyde Microcapsules

**DOI:** 10.3390/ma17040943

**Published:** 2024-02-18

**Authors:** Hongliang Zhang, Tong Yao, Fenglei Cheng

**Affiliations:** 1Key Laboratory for Special Area Highway Engineering of Ministry of Education, Chang’an University, Xi’an 710064, China; tyao@chd.edu.cn (T.Y.); chd2022021062@163.com (F.C.); 2Shandong Hi-Speed Group Co., Ltd., Jinan 250000, China

**Keywords:** bituminous mixture, microcapsule, road performance, self-healing ability, urea–formaldehyde

## Abstract

Urea–formaldehyde (UF) is a common shell material for self-healing microcapsules; however, the influence of urea–formaldehyde microcapsules (UFMs) on the road performance of bituminous mixtures and the sensitivity of their healing abilities remains unclear. In this paper, UFMs were prepared via in situ polymerization (ISP), followed by an investigation into the road performance of UFM self-healing bituminous mixtures through various tests, including wheel tracking, immersed Marshall, freeze–thaw splitting, low-temperature bending, and three-point bending fatigue tests. Subsequently, the impact of the damage degree, healing duration, and temperature on the self-healing property was discussed. The results indicated that incorporating 3 wt% UFMs into bitumen significantly improved the high-temperature stability and fatigue resistance of the bituminous mixture; for example, its dynamic stability and fatigue life could be increased by about 16.5% and 10%, respectively. However, it diminished the thermal crack resistance, as evidenced by decreases in bending tensile strength and strain by 3.7% and 10.1%, respectively. And it did not markedly improve the moisture susceptibility. Additionally, the maximum improvement observed in the healing rate was about 9%. Furthermore, the healing duration and temperature positively influenced the bituminous mixture’s self-healing, whereas the degree of damage exerted a negative impact, with a relatively significant effect.

## 1. Introduction

Under the cumulative stress of traffic loads and external environmental influences, such as water, frost, and road salt, bituminous pavements inevitably succumb to fatigue cracking. Furthermore, the infiltration of rainwater and salt water into these cracks results in looseness, potholes, and potential structural damage, thereby adversely affecting the pavement’s service life and performance [1,2]. Fortunately, bitumen has been demonstrated to possess self-healing properties, whereby microcracks in the bituminous pavement can contract or even heal during rest periods [3]. However, this self-healing process is relatively sluggish and less apparent at low temperatures and becomes less effective as bitumen ages. Aging causes the bitumen to lose volatile substances, causes it to become brittle and hard, and increases its elastic modulus, making the pavement more susceptible to cracking [4,5]. The aging process also increases the viscosity and reduces the fluidity of bitumen, further impairing its healing performance [6]. Consequently, these factors significantly challenge the ability of bitumen to counteract the damage sustained during routine usage in practical scenarios through its inherent healing capabilities alone [7,8].

To address the aforementioned challenges, numerous researchers have proposed some methods to augment the self-healing properties of bitumen, which can be divided into thermally induced and rejuvenator encapsulation-induced healing. Thermally induced healing technologies are categorized into induction heating and microwave heating [9,10,11,12]. Induction heating necessitates incorporating materials like steel wool and fibers into the bitumen layer to enhance conductivity, leading to increased costs [13]. Microwave heating was regarded as an advancement over induction heating, offering a significantly higher heating rate without necessitating additional conductive agents, or requiring only minimal amounts [14,15]. Nevertheless, thermally induced healing necessitates human intervention throughout the service life of bituminous pavements, and the heating process might expedite the aging of bitumen [16]. In response to this, researchers have proposed the method of encapsulating the rejuvenator with microcapsules or fibers [17,18]. The fiber encapsulation ensures a higher likelihood of rejuvenator release into cracks due to the increased probability of cracks intersecting the fiber network [19]. However, this method’s development is still nascent, necessitating further research to explore the impact resistance and efficacy of self-healing in fibers containing rejuvenators within bituminous materials [15,20,21,22]. Currently, the microcapsule system remains the predominant method for encapsulating rejuvenators. Within this system, the rejuvenator is encapsulated in a core–shell structure, employing techniques such as absorption wrapping (AW) [23,24], ISP [25,26], and orifice coagulation bath (OCB) [27,28]. The core materials frequently used include sunflower oil, waste oil, epoxy resin, and various commercial rejuvenators. The shell materials typically encompass calcium alginate (CA), methanol–melamine–formaldehyde (MMF), melamine–urea (MUF), and UF. Numerous studies have shown that incorporating these microcapsules into bitumen or bituminous mixtures significantly enhanced their self-repair abilities, while the influence on other properties is acceptable [27,29,30], indicating that the microcapsule encapsulation system has great development potential.

UF, synthesized primarily from urea and formaldehyde, as a commonly used microcapsule shell material; has been widely recognized for its rapid reaction speed, stability, and excellent mechanical properties; and has aroused great interest. Various studies have explored the synthesis conditions and properties of UFM. For instance, Li, Shirzad, and Katoueizadeh et al. [31,32,33] determined optimal parameters, such as the emulsion stirring speed, pH, reaction temperature, core–shell ratio, and formaldehyde-to-urea molar ratio, by analyzing UFM’s micromorphology, particle size distribution, uniformity, and quality. Li and Kosarli et al. [34,35] observed that UFM’s morphology was nearly spherical with a rough surface, featuring wrinkles and indentations, as seen through scanning electron microscopy (SEM). This morphology improved the mechanical interlocking and stability of UFM within bitumen. Differential scanning calorimetry (DSC) and thermogravimetric analysis (TGA) results revealed that UFM remained stable even at temperatures above 200 °C; thus, making it capable of enduring the mixing temperatures (160–180 °C) of bituminous mixtures [32,33,36]. In addition, only a few UFMs with excessively large particle sizes might exhibit reduced stability due to incomplete UF polymerization [34]. Several researchers have also explored the properties of bitumen containing UFMs. Regarding the three major indicators, although results varied among studies, they collectively suggested that UFM could enhance the penetration, softening point, and ductility of aged bitumen to a certain degree [35,37]. The results from dynamic shear rheometer (DSR), bending beam rheometer (BBR) tests, and multi-stress creep recovery (MSCR) tests indicated that incorporating UFMs enhanced the adhesion, elasticity, and rutting resistance of bitumen, though it might marginally diminish its thermal crack resistance [37,38]. The self-healing capacity of bitumen containing UFMs can be assessed through metrics such as ductility, strength, complex shear modulus, fatigue life, accumulative dissipated energy, and crack width in both cured and initial specimens [33,36,39]. A majority of related studies have demonstrated that UFM significantly enhanced the self-healing capabilities of bitumen, particularly for pressure aging vessel (PAV) bitumen [40]. Regarding UFM bituminous mixtures, Aguirre et al. studied the stiffness recovery of the bituminous mixture after incorporating UF and polyurethane double-shelled microcapsules via a three-point bending test and observed the change in crack width with time by optical light microscope [36].

The aforementioned literature review reveals that UFM possesses excellent properties, making it highly suitable for bituminous pavements and effectively enhancing the self-healing ability of bitumen. However, contemporary research primarily concentrated on the performance characterization of UFM, encompassing aspects such as particle size distribution, chemical composition, micromorphology, and thermal stability. Moreover, several studies have explored the high-temperature, low-temperature, rheological, and self-healing characteristics of bitumen incorporated with UFMs. However, there is a noticeable gap in research regarding the road performance, including high-temperature stability, moisture susceptibility, thermal-crack resistance, and anti-fatigue performance. Additionally, research into the factors influencing the self-healing properties of UFM bituminous mixtures is notably limited.

The primary aim of this study was to examine the road performance and self-healing property of UFM bituminous mixtures. Firstly, UFM with ZS-1 bitumen rejuvenator as the core was prepared by ISP. Then, a series of tests, including the wheel tracking test, immersed Marshall test, freeze–thaw splitting test, low-temperature bending test, and three-point bending fatigue test, were conducted to evaluate the road performance. Finally, the impact of the damage degree, healing duration, and temperature on the self-healing property was analyzed using the fatigue–healing–fatigue test.

## 2. Materials and Methods

### 2.1. Synthesis of Microcapsules

Microcapsules encapsulating ZS-1 bitumen rejuvenator were fabricated using UF as the shell material through the ISP method. The preparation process of UFM, illustrated in Figure 1, involves three steps: synthesis of UF resin prepolymer, emulsification of the core material, and synthesis of the microcapsules. For a more detailed description of the process, refer to a previous study [37]. Figure 2 depicts the particle size distribution of UFMs. The average particle size of UFMs is 100.50 µm, with the majority of particle sizes concentrated in the 60–120 µm range.

### 2.2. Fabrication of Bituminous Mixture Specimen

In this study, the AC-13 bituminous mixture was chosen, with its aggregate gradation displayed in Figure 3. The coarse aggregates, fine aggregates, and mineral powder were derived from basalt, limestone, and limestone mineral, respectively. The properties of these aggregates are detailed in Table 1, Table 2 and Table 3. The PG 58-16 matrix bitumen was chosen, and its properties are delineated in Table 4. For the preparation of self-healing bitumen, matrix bitumen was placed in a container, heated to 130–150 °C, followed by the incorporation of 3 wt% UFMs, and then the mixture was stirred at 800–1000 rpm for 15 min. The properties of the self-healing bitumen are presented in Table 4. The optimal bitumen content was determined to be 5.0% through the Marshall mix proportion design. Subsequently, prismatic, cylindrical, and compacted slab specimens of the bituminous mixture, both with and without capsules, were fabricated. The procedure began by heating the aggregate to approximately 163 °C and the bitumen to 150 °C. Following this, the specimens were compacted within a temperature range of 120–150 °C. For cylindrical specimens, compaction was achieved using the Marshall compactor, with each side receiving 75 blows (50 for immersion Marshall test specimens). Meanwhile, the compacted plate specimens were formed using the wheel compactor, typically undergoing around 24 compaction passes.

### 2.3. Wheel Tracking Test

The dynamic stability (DS) of the specimens was determined using the wheel tracking test, as per the method in ISO 12697-22 [41]. In the experiment, 300 mm × 300 mm × 50 mm compacted slab specimens of bituminous mixture were subjected to compression by a solid rubber wheel at 0.7 MPa and 60 °C, with the specimen deformations recorded at 45 min (*t*_1_) and 60 min (*t*_2_). Each test was repeated three times, totaling six bituminous mixture compacted plates, and the average value was taken as the final result.

### 2.4. Immersed Marshall Test

Conforming to ASTM D6927-15 [42], the immersion Marshall test was conducted. The specified cylindrical specimen size for this test was Ø101.6 mm × 63.5 mm, and the specimens were categorized into two groups. For the first group, the Marshall stability (*MS*) was measured after immersing the specimens in water at 60 °C for 0.5 h. For the second group, the Marshall stability (*MS*_1_) was assessed post-immersion of the specimens in water at 60 °C for 48 h. Water stability was characterized by the residual stability (*MS*_0_). Five replicates were performed for each test, totaling ten cylindrical specimens, and the final data were derived from the average.

### 2.5. Freeze–Thaw Splitting Test

Employing the test methods outlined in AASHTO T283 [43], the experiment comprised two parts: One group of specimens was immersed in water at 25 °C for 2 h, and following this, the splitting strength, δ1, was determined. Subsequently, another group of specimens was first subjected to vacuum conditions for 16 h at −18 °C, then immersed in water at 60 °C for 24 h, and finally in water at 25 °C for 2 h, after which the splitting strength, δ2, was evaluated. Water stability was characterized by the freeze–thaw splitting strength ratio, termed *TSR*. Each test was repeated three times, totaling six cylindrical specimens, and the average value was taken as the final result.

### 2.6. Low-Temperature Bending Test

Referencing the method in ASTM D790-17 [44], the low-temperature bending test was conducted on prismatic specimens measuring 250 mm × 30 mm × 35 mm. The specimens were subjected to a central load at a rate of 50 mm/min at −10 °C until failure occurred. Bending tensile strength and bending tensile strain were employed as evaluation indicators of low-temperature crack resistance. Three replicates were utilized for each test, totaling six prismatic specimens, and the final data were derived from the average.

### 2.7. Three-Point Bending Fatigue Test

The three-point bending fatigue test was performed using a universal testing machine under controlled stress, following ASTM D7460-10 [45], employing stress ratios of 0.2, 0.3, and 0.4. Various bituminous mixtures were tested using sinusoidal loading at a frequency of 10 Hz and temperature of 20 °C. The number of load repetitions until specimen failure was considered as the fatigue life, serving as a metric to evaluate the anti-fatigue performance of the bituminous mixture. Three replicates were utilized for each test, totaling eighteen prismatic specimens, and the final data were derived from the average.

### 2.8. Fatigue–Healing–Fatigue Test

The fatigue–healing–fatigue test process encompassed three stages: initial loading, resting, and final loading. The loading mode was the same as the three-point bending fatigue test, with a stress ratio of 0.2. The self-healing property of various specimens was assessed based on the healing rate (*P*), as delineated in Equation (1) [46]. To investigate the factors influencing self-healing, the healing rates of specimens under varying conditions of damage (10%, 20%, and 30%), healing duration (6 h, 12 h, and 24 h), and temperature (20 °C, 40 °C, and 60 °C) were studied. Each test was repeated three times, totaling forty-five prismatic specimens, and the corresponding average was taken as the final result. For a clear description of the test process, a flowchart is plotted in Figure 4.
(1)$$P=(N-N_{f})/N_{f}$$
where *N* is the sum of loading times before and after the resting of bituminous mixture containing UFMs; and *N_f_* is the fatigue life of matrix bituminous mixture.

## 3. Results and Discussion

### 3.1. High-Temperature Stability

The wheel tracking test was performed on bituminous-mixture slabs to evaluate the high-temperature performance of mixtures containing UFMs, and the results are illustrated in Figure 5. It was seen that the deformation of specimens with UFMs was slightly higher than that of the matrix bituminous mixture at 45 min. This observation could be attributed to a small number of UFMs on the specimen surface breaking during the wheel-running period, with the leaked core material potentially aiding in further compacting the aggregate and reducing the mixture’s internal friction through lubrication, consequently diminishing the surface shear strength [47]. Fortunately, the deformations of both specimen types were comparable at 60 min, indicating that the UFM maintained good mechanical properties and did not continue breaking. Overall, the dynamic stability of the UFM bituminous mixture specimen is approximately 16.5% higher than that of the matrix bituminous mixture specimen, demonstrating that UFM incorporation enhances the high-temperature anti-rutting performance of the bituminous mixture. This phenomenon could be attributed to the fact that the UFM existed in the form of particles in the bitumen and might act as a reinforcing filler, resulting in a minimal loss of elasticity, which led to an increase in viscosity and the rutting factor of the bitumen [35]. Simultaneously, some aromatic and saturated components in the bituminous material were absorbed by the UFM, leading to the hardening of the bitumen. These two reasons resulted in an augmented deformation resistance of the bituminous mixture.

### 3.2. Moisture Susceptibility

Moisture susceptibility was assessed via the immersed Marshall test and the freeze–thaw splitting test, and the corresponding results are presented in Figure 6 and Figure 7. It can be observed that the *MS*_0_ and *TSR* results demonstrate exceptional consistency; that is, the moisture susceptibility of the bituminous mixture containing UFMs is slightly higher than that of the matrix bituminous mixture. For instance, the *MS*_0_ of the UFM bituminous mixture increased by 2.27% compared to that of the matrix bituminous mixture. Possible reasons for this phenomenon include the following: firstly, the rough surface of UFM, as previously mentioned, might create a locking effect with the aggregate, thereby enhancing the adhesion between the bitumen and aggregate [48]; secondly, the selection of hydrophilic UF as the capsule shell material (due to the capsule core being lipophilic) in this paper, which absorbed some of the infiltrated water, potentially mitigating water damage to the specimen [49].

### 3.3. Thermal Crack Resistance

The results of the thermal crack resistance for bituminous mixtures with and without microcapsules are displayed in Figure 8. It can be found that the incorporation of UFMs negatively influences the thermal crack resistance of bituminous mixtures, with decreases in bending tensile strength and strain by 3.7% and 10.1%, respectively. The explanation for these phenomena was that the incorporation of UFMs disrupted the homogeneous structure of the bitumen, reducing its flexibility and ductility. Additionally, the swelling of microcapsule particles led to volume expansion, which tended to produce greater temperature stress, consequently diminishing the thermal crack resistance of the bituminous mixture. However, these results still met the requirements for matrix bituminous mixtures in 1–3, 2–3, 1–4, and 1–4 climatic zonation in China, stipulating that the bending tensile strain should be no less than 2000 με.

### 3.4. Anti-Fatigue Performance

Table 5 exhibits the failure stresses of bituminous mixtures with and without microcapsules. It is observed that the failure stress of the bituminous mixture containing UFMs is slightly higher than that of the matrix bituminous mixture. Table 6 presents the fatigue test results of bituminous mixtures with and without microcapsules under differing stress ratios. Clearly, as the stress ratio increases, the fatigue life of both bituminous mixtures decreases gradually. Furthermore, the fatigue life of the UFM bituminous mixture surpasses that of the matrix bituminous mixture at the same stress ratios, particularly at 0.4, indicating a superior fatigue resistance of the former. This phenomenon could be primarily attributed to the fact that the incorporation of UFMs hardened the bitumen, as outlined in Section 3.1. It is widely recognized that the fatigue life of a bituminous mixture increases with the hardening of bitumen under stress-controlled loading conditions.

Numerous studies demonstrate a linear relationship between stress (*σ*) and fatigue life (*N_f_*) for different stress ratios in double logarithmic coordinates, as shown in the Equation (2).
(2)$$\mathrm{lg}N_{f}=K-n\mathrm{lg}\sigma$$
where *K* and *n* denote the regression constants related to the material properties and test conditions, respectively. The fatigue regression curves of bituminous mixtures with and without microcapsules are depicted in Figure 9, where the intercept characterizes the anti-fatigue performance, and the slope indicates the sensitivity of the *N_f_* value to the stress level [50]. It can also be seen from the slope of the fatigue regression curve that the UFM bituminous mixture exhibits better fatigue resistance. Furthermore, observation reveals that the slope of the regression curves for UFM bituminous mixture and matrix bituminous mixture are nearly identical, indicating a similar level of sensitivity to stress levels.

### 3.5. Self-Healing Performance

In this paper, the impacts of the damage degree, healing duration, and temperature on the self-healing property of bituminous mixtures with and without microcapsules were studied by fatigue–healing–fatigue test, with the healing rate as the evaluation index.

#### 3.5.1. Effect of the Damage Degree on Healing Rate

The damage degree is directly correlated with the width and density of the crack and constitutes an important factor influencing the healing rate. In this paper, the damage degree was characterized by the number of initial loading cycles. For instance, with 10% damage, the number of times of initial loading is 10% of the fatigue life when the stress ratio is 0.2, equating to 1050 cycles. Following the initial loading, the specimens were allowed to rest for 24 h at 20 °C, subsequently undergoing final loading until failure. Figure 10 illustrates the variation in the healing rate with damage degree for bituminous mixtures with and without microcapsules. It is notable that the healing rate significantly decreases with an increasing degree of damage, regardless of bituminous mixture type. For example, when the damage level escalates from 10% to 30%, the healing rate of the UFM bituminous mixture drops from 27.02% to 7.23%. This phenomenon could be explained by the critical point of damage in the bitumen mixture specimens. Below this critical point, the material exhibited microcracks and required less self-healing energy, while beyond this point, specimens with high damage levels tended to exhibit large macrocracks, demanding a significant amount of self-healing energy, or in some cases, making self-healing challenging [51]. Additionally, the healing rate of the UFM bituminous mixture is consistently higher than that of the matrix bituminous mixture at different damage degrees, indicating that the rejuvenator released from the broken microcapsules effectively softened the bitumen and promoted the self-healing of cracks, thereby extending the fatigue life of the specimens.

#### 3.5.2. Effect of the Healing Duration on Healing Rate

Bitumen exhibits viscoelastic polymeric properties with time–temperature-dependent behavior. In order to investigate the impact of the healing duration on healing rate, specimens with a 10% damage degree were conditioned at 20 °C for 6, 12, and 24 h to rest, followed by the final loading until failure. Figure 11 depicts the variation in healing rate with respect to healing duration for bituminous mixtures with and without microcapsules. It is evident that the healing rate of the mixture increases over time, and upon reaching a healing duration of 12 h, the increasing speed in healing rate decreases. This phenomenon indicates that, following the appearance of cracks, sufficient time was necessary for the specimens to allow for the rejuvenator released from the microcapsules to fill the cracks and facilitated the self-healing of the bituminous mixture. However, beyond a healing duration of 12 h, self-healing was basically completed, and the benefit of prolonging healing duration on the healing rate began to decrease. In addition, it can be found that the healing rate of the UFM bituminous mixture is always higher than that of the matrix bituminous mixture at different healing durations, thus showing the effectiveness of UFM in improving the self-healing property of bituminous mixtures.

#### 3.5.3. Effect of the Healing Temperature on Healing Rate

Specimens with 10% damage were subjected to resting periods at 20 °C, 40 °C, and 60 °C for 6 h to investigate the effect of temperature on healing rate. Figure 12 illustrates the variation in healing rate with temperature for bituminous mixtures with and without microcapsules. It is evident that both UFM and matrix bituminous mixtures exhibit a continuous linear increase in the healing rate with increasing temperature, demonstrating the significant impact of temperature on the healing rate. For instance, with the UFM bituminous mixture, as the healing temperature rises from 20 °C to 40 °C and subsequently to 60 °C, the healing rate escalates from 11.78% to 15.69%, and further to 19.92%. The underlying reason for this phenomenon was that the bitumen softened, and its viscosity reduced as the temperature rose in the matrix bituminous mixture, thereby enhancing its self-healing capability. As for UFM bitumen mixture, in addition to this reason, the increase in temperature also increased the diffusion rate of the rejuvenator, allowing it to penetrate the bitumen more quickly [52]. Furthermore, under different healing times, the healing rate of UFM bituminous mixtures consistently surpassed that of matrix bituminous mixtures, underscoring the efficacy of UFM in enhancing the self-healing property of the bituminous mixture.

A comparison of Figure 10, Figure 11 and Figure 12 clearly indicates that the degree of damage exerts the greatest influence on the healing rate, followed by healing duration and temperature. When the damage degree reaches 30%, the enhancement in the healing rate achieved through the introduction of UFMs became constrained.

## 4. Conclusions

In this paper, comparative research on the road performance of matrix bituminous mixtures and UFM bituminous mixtures was implemented, and the factors that affect the self-healing ability were analyzed. Based on the experimental studies and data analysis, the following conclusions can be drawn:

1.The incorporation of UFMs enhancing the high-temperature rutting resistance of the bituminous mixture and yielding a 16.5% increase in DS.2.Compared to the matrix bituminous mixture, the UFM bituminous mixture exhibited a 2.27% increase in *MS*_0_ and a 2.95% increase in *TSR*, indicating a slight improvement in moisture susceptibility.3.UFM exhibited a negative effect on the thermal crack resistance of the bituminous mixture, with reductions in bending tensile strength and strain by 3.7% and 10.1%, respectively; however, it still met the specifications of China.4.The fatigue life of the UFM bituminous mixture was higher than that of the matrix bituminous mixture at all stress levels, particularly at a stress ratio of 0.4; however, both were about equally sensitive to stress levels.5.The introduction of 3 wt% UFMs significantly enhanced the self-healing ability of the matrix bituminous mixture, and the healing rate could increase by about 9% at most. Additionally, as the damage degree increased, the healing rate of bituminous mixtures with and without microcapsules noticeably decreased, while elevating the healing temperature and duration could enhance the healing rate.

## Figures and Tables

**Figure 1 materials-17-00943-f001:**
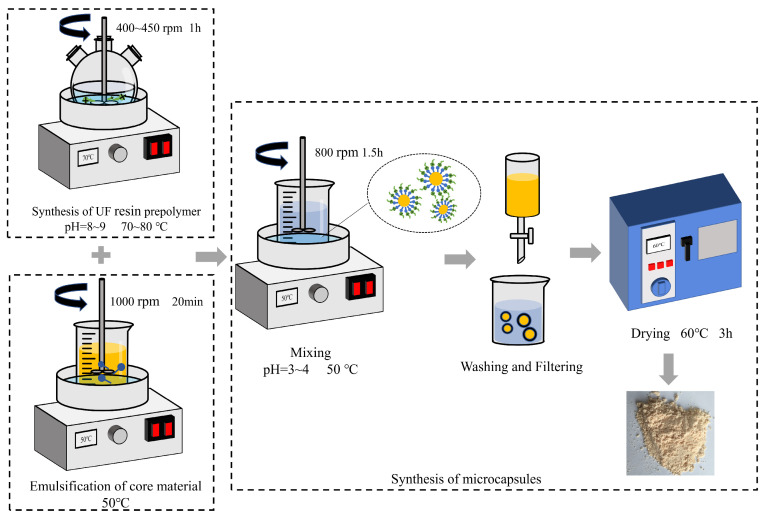
Preparation process of UFMs.

**Figure 2 materials-17-00943-f002:**
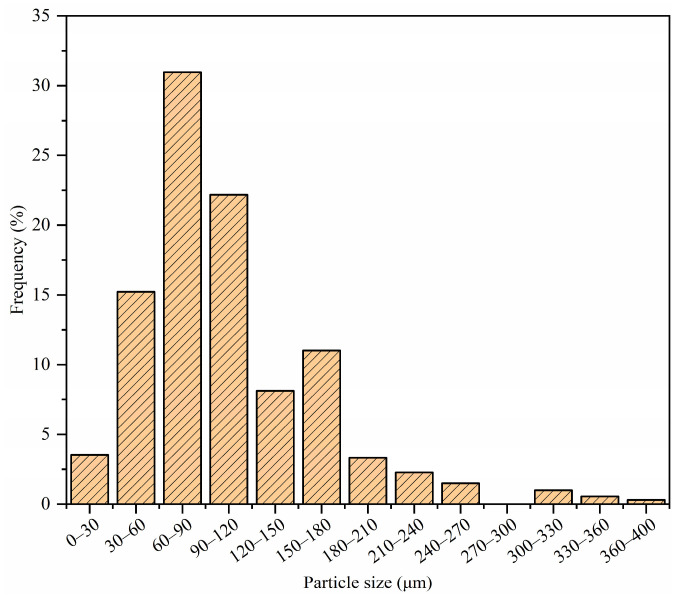
Particle size distribution of UFMs.

**Figure 3 materials-17-00943-f003:**
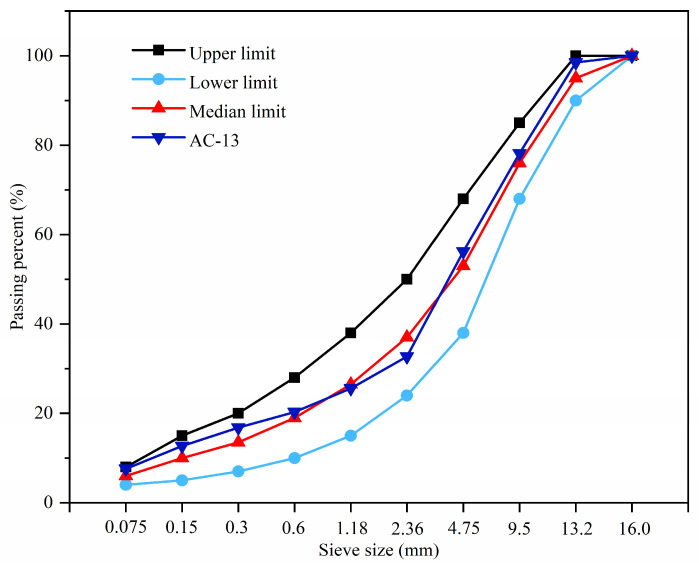
Particle size distribution of the aggregate.

**Figure 4 materials-17-00943-f004:**
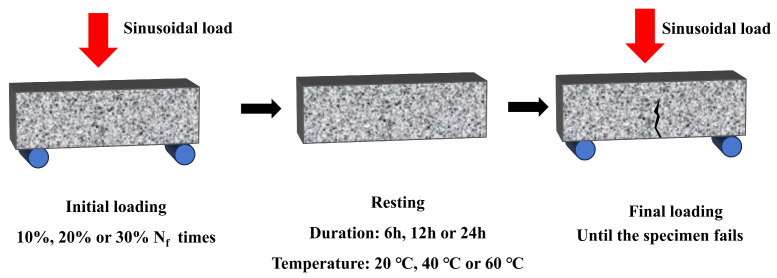
Flowchart of fatigue–healing–fatigue test process.

**Figure 5 materials-17-00943-f005:**
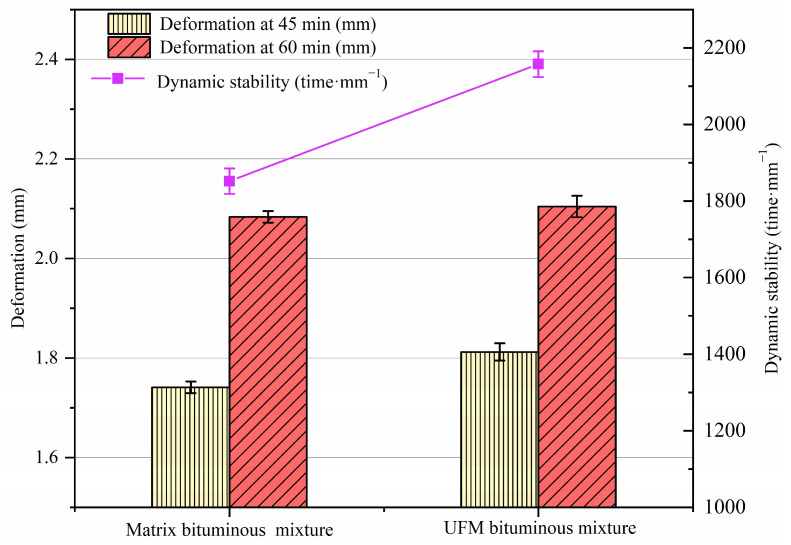
Wheel tracking test results of bituminous mixtures specimens with and without microcapsules.

**Figure 6 materials-17-00943-f006:**
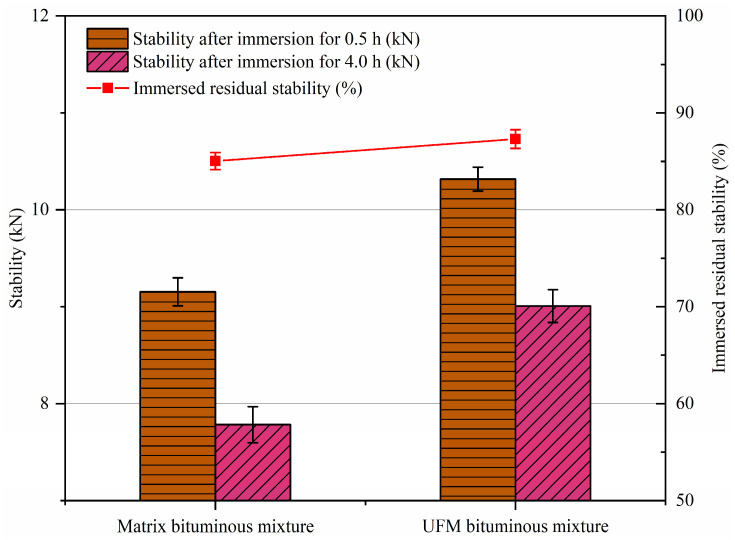
Immersed Marshall test results of bituminous mixtures specimens with and without microcapsules.

**Figure 7 materials-17-00943-f007:**
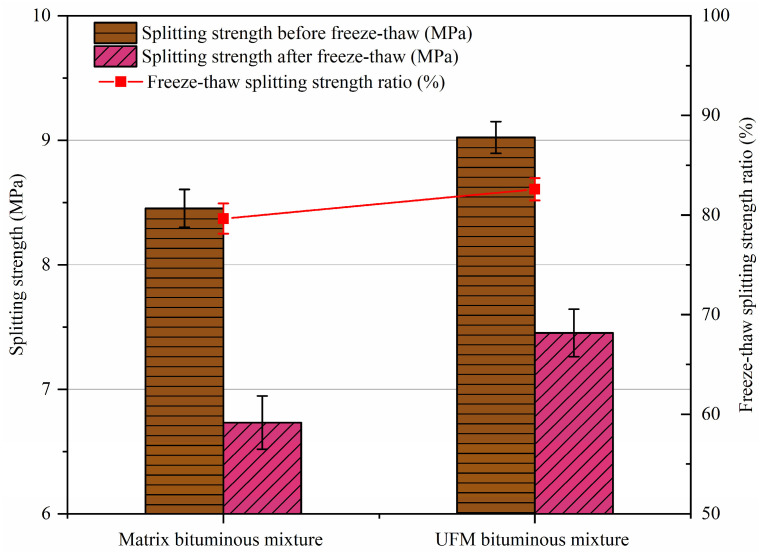
Freeze–thaw splitting test results of bituminous mixtures specimens with and without microcapsules.

**Figure 8 materials-17-00943-f008:**
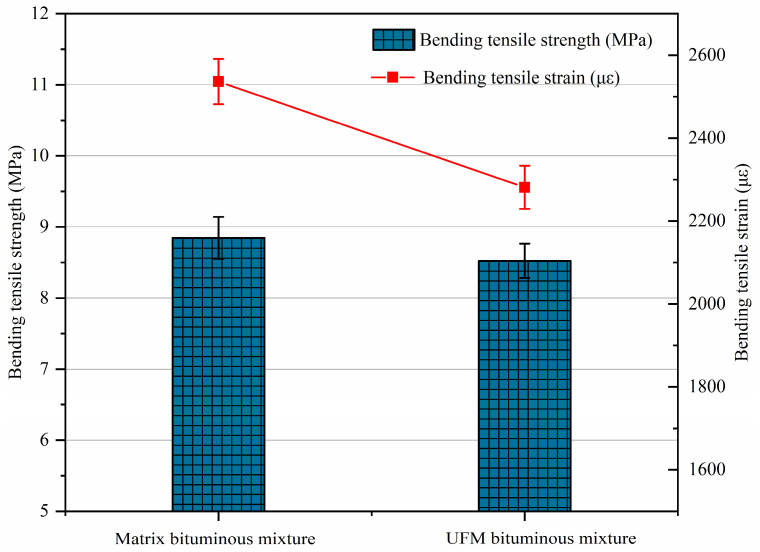
Low-temperature bending test results of bituminous mixtures specimens with and without microcapsules.

**Figure 9 materials-17-00943-f009:**
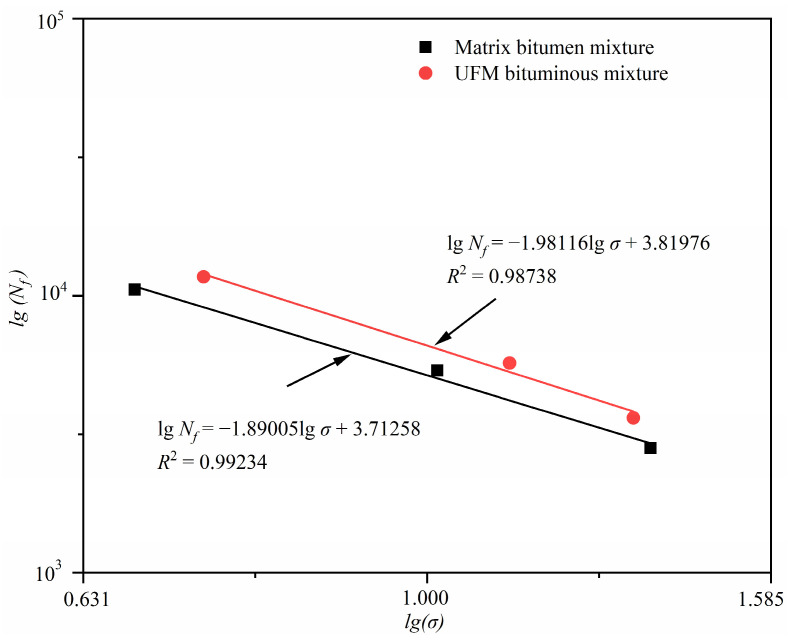
Fatigue regression curves of bituminous mixtures with and without microcapsules.

**Figure 10 materials-17-00943-f010:**
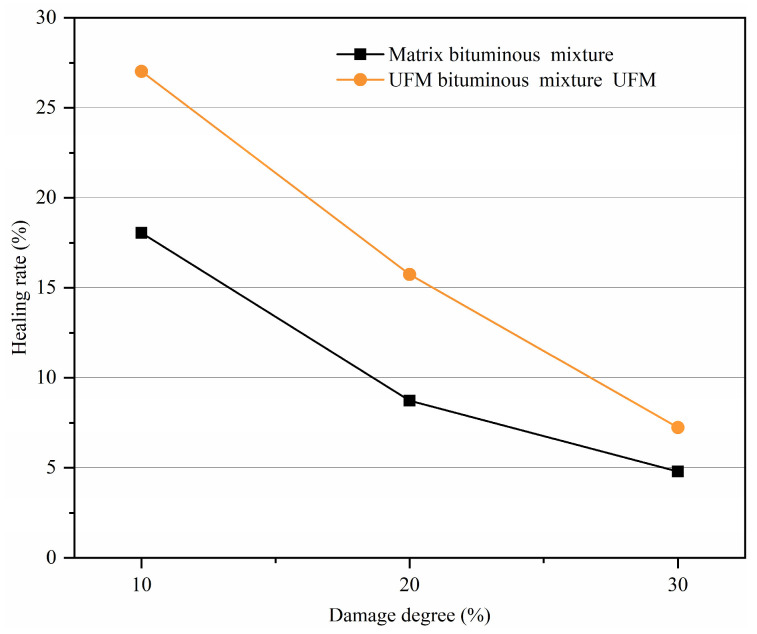
Variation in healing rate with damage degree for bituminous mixtures with and without microcapsules.

**Figure 11 materials-17-00943-f011:**
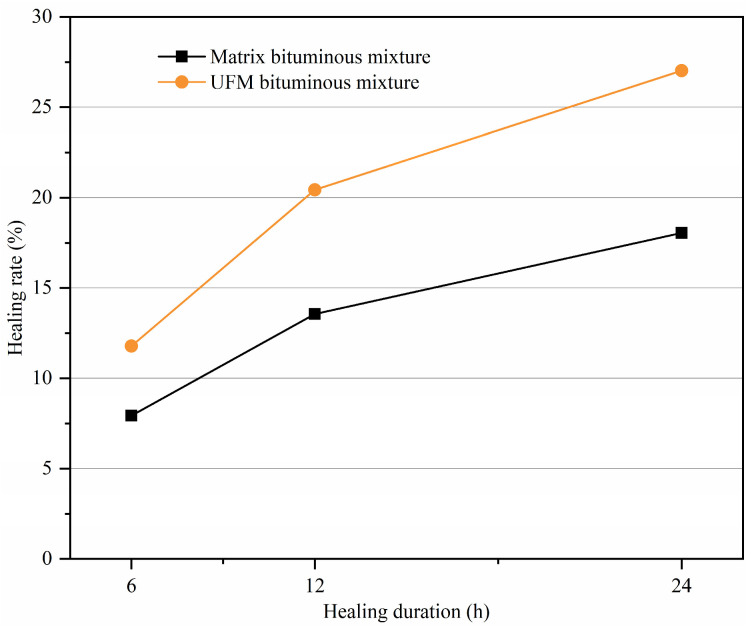
Variation in healing rate with healing duration for bituminous mixtures with and without microcapsules.

**Figure 12 materials-17-00943-f012:**
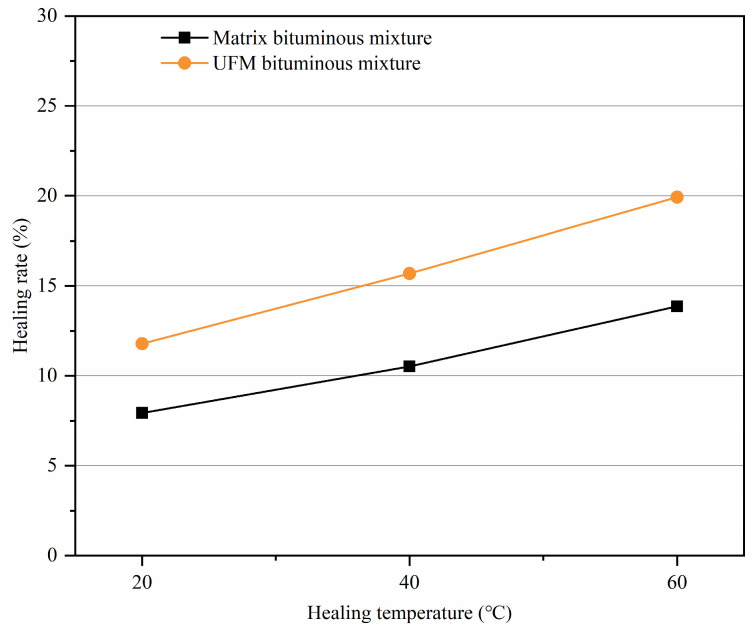
Variation in healing rate with healing temperature for bituminous mixtures with and without microcapsules.

**Table 1 materials-17-00943-t001:** Properties of coarse aggregate.

Properties		Results	Standard Requirements
Apparent relative density	9.5–16 mm	2.923	≥2.60
	4.75–9.5 mm	2.926	
	2.36–4.75 mm	2.896	
Water Absorption (%)	9.5–16 mm	0.5	≤2.0
	4.75–9.5 mm	0.64	
	2.36–4.75 mm	0.53	
Crushing value (%)		11.8	≤26
Polish stone value (PSV)		43	≥42
Soft rock content (%)		0.71	≤3
Los Angeles abrasion loss (%)		10.6	≤28

**Table 2 materials-17-00943-t002:** Properties of fine aggregate.

Properties	Results	Standard Requirements
Apparent relative density	2.679	≥2.50
Sand equivalent (%)	64	≥60
Angularity (S)	42.7	≥30
Mud content (<0.075 mm) (%)	2.1	≤ 3

**Table 3 materials-17-00943-t003:** Properties of mineral power.

Properties		Results	Standard Requirements
Apparent relative density		2.761	≥2.50
Water content (%)		0.4	≤1
Particle size range (%)	<0.6 mm	100	100
	<0.15 mm	92.9	90~100
	<0.075 mm	81.8	75~100
Hydrophilic coefficient		0.70	<1

**Table 4 materials-17-00943-t004:** Properties of bitumen.

Bitumen	Matrix Bitumen	Bitumen Containing 3 wt% Microcapsules
Penetration (25 °C, 100 g, 5 s) (0.1 mm)	90.8	73.3
Softening point (°C)	47.3	52.4
Ductility (50 mm/min, 10 °C) (cm)	43.8	26.5

**Table 5 materials-17-00943-t005:** Failure stress of bituminous mixtures with and without microcapsules.

Types	Matrix Bituminous Mixture	UFM Bituminous Mixture
Failure stress (MPa)	3.38	3.72

**Table 6 materials-17-00943-t006:** Fatigue test results of bituminous mixtures under different stress ratios.

Types	Stress Ratio	Stress, *σ* (MPa)	lg (*σ*)	Mean of Fatigue Life, *N_f_* (Number of Times)	lg (*N_f_*)
Matrix bituminous mixture	0.2	0.676	−0.17	10,524	4.022
	0.3	1.014	0.006	5465	3.73
	0.4	1.352	0.13	2847	3.45
UFM bituminous mixture	0.2	0.744	−0.13	11,697	4.068
	0.3	1.116	0.048	5709	3.757
	0.4	1.336	0.12	3627	3.559

## Data Availability

The data presented in this study are available upon request from the corresponding author.

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
