# Peer review of "Road Performance and Self-Healing Property of Bituminous Mixture Containing Urea–Formaldehyde Microcapsules"

_materials, 2024, doi:10.3390/ma17040943_

Round 1
Reviewer 1 Report
Comments and Suggestions for Authors
A very interesting article presents interesting issues of self-healing.
A large number of studies were included and the results were well reported.
Additionally, the following publication should be included in the review of the state of knowledge:
https://doi.org/10.3390/ma16030966
regarding the processes of the interaction of salt and atmospheric factors.
Also correct Figure 10. Present the value of loads N on the vertical axis and stresses on the horizontal axis.
Do not use a logarithmic scale. Let the graphs be logarithmic functions of the form N = f(log σ). You can also reverse the axes but keep the formula in the appropriate form: σ = f(log N)
Reviewer 2 Report
Comments and Suggestions for Authors
I have found this paper is well-presented and have a potential to influence the researchers from other research and industry fields. The subject of repairing deteriorated road pavements taking into account the aging processes of asphalt is topical. The authors examined the influence of air voids on the process of rutting damage. The subject matter is within the scope of the journal. The results are clear. The work is well written but needs some adjustments. Some suggestions for improvement are given as follows:
Abstract - Add some of the most critical quantitative results to the Abstract.
Introduction - The authors rightly noted that the self-healing property of asphalt pavements is affected by the aging processes of bitumen. Therefore, it is necessary in the article to also introduce the subject of aging of asphalt. Please refer to the following articles in this area: Understanding the bitumen ageing phenomenon: A review, doi: 10.1016/j.conbuildmat.2018.10.169; Modern two-component modifiers inhibiting the aging process of road bitumen 10.1016/j.conbuildmat.2023.133838,.
Materials and Methods
Complete the information on process temperatures: asphalt and aggregate temperature, sample compaction temperature and other parameters during sample compaction. Complete the data on the number of samples tested.
Descriptions of standard test methods should be shortened. Please provide the main data adopted by the authors, while the formulas for calculations are in the standards.
Conclusions
Final conclusions should follow from the study. The authors did not prove that UFM sorbed light components of asphalt.
I hope authors can consider improving the manuscript accordingly. I hope that the feedback I provided should be useful for authors.

Reviewer 3 Report
Comments and Suggestions for Authors
Comments of this reviewer on the manuscript Materials-2856166 are as follows:
1. To the best of this reviewer's knowledge, this manuscript has been prepared properly and has all the necessary parts, namely: modeling, theory, experimental background, results, discussion of results, and conclusions that have scientific significance.
2. There are a few minor comments of this reviewer. They are as follows.
- Keywords should be listed in alphabetical order.
- The introduction is too long and contains some uncommon details. For instance: Figure 1 should be excluded from the introduction together with the associated description, the literature review should be shortened, etc.
- In the introduction, one can find the following sentence: “Nevertheless, the World Health Organization (WHO) noted that the human body could absorb microwave radiation, potentially leading to the heating of exposed tissues and subsequent thermal injuries [11].” In this regard, the opinion of this reviewer is that the World Health Organization and comparisons with medicine should be excluded from manuscripts of this type. Of course, the given sentence can be found in Reference [11] as well.
- Table 4 is divided into two parts that are provided on different pages.
- Equation appearing on Page 10 lacks the corresponding ordinal number (7).
- The first author of this manuscript contributed the following publication: https://www.sciencedirect.com/science/article/abs/pii/B9780128189818000096, which deals with the similar topic. This publication was not cited in this manuscript. Why? Accordingly, the results obtained in this manuscript must be contrasted to the ones from https://doi.org/10.1016/B978-0-12-818981-8.00009-6.
Reviewer 4 Report
Comments and Suggestions for Authors
Dear Authors, this article is very important for academic and engineering auditorium globally and locally, dealing with similar issues on roads construction and maintenance. Some comments about possible improvements to this research manuscript are provided below.
1. By the end of Introduction some details of novelty about this research can be clearly defined, instead of general statement “…the investigation into UFM bituminous mixture is still considerably lacking, particularly in terms of its road performance “.
2. Figure 1 is not of scientific origin, and it is more suitable to publish in graphical abstract section.
3. Figures 12 and 13 can be combined in one and explained with comparison research results obtained using different and the same experimental admixtures.
4. ISO standards can be added to Conclusion 3, to attract the attention of international readers and compare similar research results obtained globally.
5. Conclusion 5 can be expressed with additional percentage or some proportional achievements using new admixtures.
6. Finally References 37, 38 and 45 can be expressed with additional ISO standards requirements to ensure globally recognised achievements.
Sincerely, Reviewer.
